



# Rapid mineralization of biogenic volatile organic compounds in temperate and Arctic soils

Christian Nyrop Albers[1,2], Magnus Kramshøj[1,2,3], Riikka Rinnan[2,3]

[1]Geological Survey of Denmark and Greenland (GEUS), Department of Geochemistry, Øster Voldgade 10, DK-1350 Copenhagen K, Denmark

[2]Center for Permafrost (CENPERM), Department of Geosciences and Natural Resource Management, University of Copenhagen, Øster Voldgade 10, DK-1350 Copenhagen K, Denmark

[3]Department of Biology, University of Copenhagen, Universitetsparken 15, DK-2100 Copenhagen E, Denmark

*Correspondence to:* Christian N Albers (cal@geus.dk)

**Abstract.** Biogenic volatile organic compounds (BVOCs) are produced by all life forms. Their release into the atmosphere is important with regards to a number of physical and chemical processes and great effort has been put into determining sources and sinks of these compounds in recent years. Soil microbes as a possible sink for BVOCs in the atmosphere has been suggested, however, experimental evidence for this sink is scarce despite its potentially high

importance to both carbon cycling and atmospheric concentrations of these gases. We therefore conducted a study with a number of commonly occurring BVOCs labelled with $^{14}$C and modified existing methods to study mineralization of these compounds to $^{14}CO_2$ in four different top soils. Five of the six BVOCs were rapidly mineralized by microbes in all soils. However, great differences were observed with regards to speed of mineralization, extent of mineralization and variation between soil types. Methanol, benzaldehyde, acetophenone and the oxygenated monoterpene geraniol were

mineralized within hours in all soils. The hydrocarbon monoterpene p-cymene was mineralized rapidly in soil from a coniferous forest but slower in soil from and adjacent beech stand while chloroform was mineralized slowly in all soils. From our study it is clear that soil microbes are able to degrade completely BVOCs released by aboveground vegetation as well as BVOCs released by soil microbes and plant roots. In addition to the possible atmospheric implications of this degradation the very fast mineralization rates are likely important in shaping the net BVOC emissions from soil and it is

possible that BVOC formation and degradation may be an important but little recognized part of internal carbon cycling in soil.



## 1 Introduction

Non-methane biogenic volatile organic compounds (BVOCs) are produced by all life forms, with plants being the most
important contributors to the atmospheric concentrations of BVOCs and also the most studied group of BVOC emitters
(Laothawornkitkul et al., 2009; Peñuelas et al., 2014). Production of BVOCs in soil (McNeal and Herbert, 2009;
Ramirez et al., 2010) and by isolated soil microorganisms (Insam and Seewald, 2010; Garbeva et al., 2014) has been
shown as well, though.

BVOCs comprise a very high diversity concerning molecular size and chemical structures, which leads to a high
compound-to-compound variation in life times and reactions in the environment. Chemical oxidation reactions are
regarded as the dominant BVOC sink in air, with impacts on the concentrations of methane, ozone, secondary organic
aerosol and even clouds (Peñuelas and Staudt, 2010; Glasius and Goldstein, 2016). In addition to chemical reactions in
the atmosphere, an uptake or deposition of BVOCs into or onto soil has been observed (Ramirez et al., 2010; Spielmann
et al., 2017) and so has a bidirectional atmosphere/soil exchange of certain BVOCs (Asensio et al., 2007; Asensio et al.,
2008; Gray et al., 2014). The mechanism behind the soil uptake has not been investigated, but it may owe to processes
like adsorption to organic matter, dissolution in soil water and microbial degradation. Adsorption and dissolution may
be predicted if the chemical characteristics of the BVOCs in question are known, and fate models for these parameters
could then be set up. The microbial degradability of BVOCs - and especially the rate of degradation - are on the other
hand very difficult to predict.

It is known from lab experiments, that many BVOCs can be degraded by soil bacteria functioning as substrates for
growth (e.g. Cripps, 1975; Misra et al., 1996; Kleinheinz, 1999; El Khawand et al., 2016). However, the studies on
BVOC degradation in soil or by isolated soil microorganisms, have typically used BVOC concentrations of 3-6 orders
of magnitude higher than those present in the environment. Degradation experiments with such high concentrations are
very well suited for selectively enriching BVOC degraders and showing the potential for use of a degrader organism in
industrial processes. However, they do not serve to assess degradation at realistic environmental concentrations that
would be too low to sustain bacterial growth singly due to degradation of a specific BVOC. Thus, we do not know if
microbial BVOC degradation in soil is of environmental importance. An exception from this is isoprene degradation, of
which there is substantial evidence at environmental conditions (Cleveland and Yavitt, 1997; Gray et al., 2015).

Degradation experiments with BVOCs in soil are difficult to interpret as the same compounds may be produced and
released by the soil while also being degraded. By using isotopically labelled compounds in degradation experiments it
is possible to target degradation alone. Isotopic labelling also enables working with compounds at lower concentrations.
This is especially true for using radioactive $^{14}$C-labelling, which furthermore enables one to determine complete
mineralization to $^{14}CO_2$. Compared to compound removal over time, complete mineralization is the ultimate proof of
degradation and is often used in pesticide fate studies. However, apart from three studies looking at mineralization of
$^{14}$C-labelled geraniol (Owen et al., 2007), methanol (Stacheter et al., 2013) and chloroform (Albers et al., 2011), we are
unaware of such studies with BVOCs. Furthermore, so far no BVOC mineralization studies were done at concentrations
observed in natural environments.





The aim of this study was to assess microbial mineralization of different BVOCs in soils from contrasting environments. The microbial sink of BVOCs in soil would be of potentially high importance to both carbon cycling and atmospheric concentrations of these gases. We therefore purchased a number of commonly occurring BVOCs labelled

with $^{14}$C and modified existing methods to study mineralization of these compounds to $^{14}CO_2$ in four different soils.

## 2 Materials and Methods

### 2.1 Soil sampling and characterization

Soil was sampled at four sites representing common ecosystem types in the temperate and Arctic temperature zones.

From the temperate zone, we sampled a coniferous forest site (12°03'40" E, 56°02'22" N) dominated by Norway spruce (*Picea abies*) and Scots pine (*Pinus sylvestris*) and a European beech (*Fagus sylvatica*) forest site (12°04'22" E, 56°02'22" N). The two sites were located 750 m apart. At both sites the Aeolian sandy soil is 200 to 400 years old and has been forested for at least 150 years. Both sites lack underwood, forest floor vegetation and moss cover, and a 5-10 cm thick organic layer has accumulated on top of the sand. Loose litter was removed before sampling the organic layer.

From the Arctic, we sampled a  tundra heath site (53°27'48" W, 69°15'49") dominated by 5-15 cm tall dwarf shrubs *Empetrum nigrum*, *Betula nana* and *Cassiope tetragona*. A 4-8 cm thick organic layer has accumulated on top of a sandy parent soil. We sampled the organic layer in between individual plants. The second Arctic site was an area with bare ground without vegetation and with coarse soil particles ("Arctic bare soil", 53°27'58" W, 69°15'57"), located 300 m from the heath site. Here, the top 5 cm was sampled.


From each site, 10-12 replicate samples were cored with a brass core (diameter 38 mm) from within a 25 m$^2$ area and pooled in a plastic bag. After arrival to the laboratory, the pooled samples were gently mixed by hand and larger roots were removed to get the final soil sample. The Arctic bare soil contained no roots and instead of mixing by hand, this soil was homogenized by sieving (5 mm). The mixed samples were stored at 3°C for a period of up to six weeks before

mineralization experiments were initiated.

Water content was determined gravimetrically after drying at 105°C for 24 hours. Soil organic matter was determined as loss on ignition (LOI; 550°C, 2h). pH was determined with a pH electrode in slurries of soil:water (1:2.5) after 30 min shaking.


For each soil, triplicate DNA extractions were made from 0.25 g subsamples using the PowerLyzer PowerSoil DNA Isolation Kit (MoBio, Carlsbad, California). Total bacterial biomass was quantified as 16s gene copies by qPCR targeting the 16S rRNA sequence using forward primer 341F (5´-CCTACGGGAGGCAGCAG-3´) and reverse primer 518R (5´-ATTACCGCGGCTGCTGG-3´) and 1 µL DNA template, as previously described (Feld et al., 2016). Total

fungal biomass was determined as ITS2 gene copies by targeting the fungal ITS2 nuclear ribosomal DNA region using




forward primer gITS7 (GTGARTCATCGARTCTTTG) and reverse primer ITS4 (TCCTCCGCTTATTGATATGC) as previously described (Christiansen et al., 2017). All qPCR was run in technical triplicates.

### 2.2 Atmospheric BVOC-concentrations

A snap-shot of the atmospheric concentration of a range of BVOCs was determined on the day of soil sampling in each of the two forest sites and in the Arctic sampling area. Triplicate 6 L air samples (12 L at the Arctic sites) were drawn through a sorbent cartridge 10 cm above soil surface (Coniferous, Beech and Arctic Bare sites) or 5 cm above the canopy (Arctic Heath site). Two types of sorbent cartridges were used in order to capture a range of BVOCs (Tenax TA/Carbograph 1TD as sorbent) and (Carbotrap B/Carboxen 1000/Carboxen 1003) to capture halogenated VOCs,

including the model compound chloroform. The sorbent cartridges were sampled and analyzed by GC-MS as previously described (Kramshøj et al., 2015; Johnsen et al., 2016)

### 2.3 Incubations for BVOC mineralization

    Six $^{14}$C-labeled BVOCs were used as model compounds representing different molecular weights and chemical classes

(Fig. 1, Table 2). $^{14}$C-methanol (58 mCi millimole$^{-1}$), [ring-$^{14}$C-]-benzaldehyde (>99% radiochemical purity; 60 mCi millimole$^{-1}$) (trans)-[1-14C]-Geraniol (99% radiochemical purity; 55 mCi millimole$^{-1}$), [ring-$^{14}$C-]-acetophenone (99% radiochemical purity; 55 mCi millimole$^{-1}$) and $^{14}$C-chloroform (>99% radiochemical purity; 2.25 mCi millimole$^{-1}$) were purchased from American Radiolabeled Chemicals Inc. (St. Louis, MO). [1-methyl-$^{14}$C]-p-cymene (96% radiochemical purity; 57 mCi millimole$^{-1}$) was purchased from Moravek (Brea, Ca). Stock solutions ($3 \times 10^7$ DPM mL$^{-1}$) were made in

sterile water (methanol and benzaldehyde), ethanol (acetophenone, geraniol and p-cymene) or acetonitrile (chloroform) and stored at -18°C until use.





**Figure 1.** Chemical structures of the used model compounds. Radiolabeled C is marked with an asterisk.


Incubations were carried out in 120 mL serum flasks. Into each flask, 5 (coniferous), 6 (beech and Arctic heath) or 10 (Arctic bare) g fresh weight (f.w.) soil with natural moisture was weighed and equilibrated overnight at 10°C. A small glass vial containing 2.5 mL 1M NaOH and 0.01M $NaHCO_3$ was placed in the flask to trap $^{14}CO_2$ liberated from $^{14}$C-BVOC mineralization. The $NaHCO_3$ was used in order to precipitate all trapped $^{14}CO_2$ with $Ba^{2+}$ added during the

following analysis procedure. 0.5 mL radiolabeled BVOCs dissolved in sterile water was then distributed across the soil with a pipette. The transfer of the BVOC had to be carried out fast in order not to loose it from the aqueous solution. For each BVOC, portions of the aqueous solution were transferred to scintillation vials containing HiSafe 3 liquid scintillation cocktail (Perkin Elmer, Waltham, MA) just before transferring to the first incubation flask and just after transferring to the last incubation flask to assure that all flasks had received similar $^{14}$C-BVOC-concentrations. The

scintillation vials were then counted on a liquid scintillation counter (Tri-Carb 2810 TR, PerkinElmer, Waltham, MA) for 30 minutes or until 1% uncertainty (2S, 95% CL) was achieved. The BVOC concentrations used for incubation corresponded to 43-73 ppbv (64-504 ng $L^{-1}$), assuming all BVOC was present in the headspace of the flasks. Most of the BVOCs were, however likely dissolved in water or adsorbed to the soil, so recalculating to a soil basis (0.8-11 µg $kg^{-1}$ f.w. soil) may be more appropriate.


Immediately after the transfer of the BVOC solution, flasks were closed with crimp-caps containing an alumina-coated septum (Mikrolab Aarhus, Denmark) and incubated at 10°C in the dark. At several time points, the alkaline $CO_2$-trap was exchanged through a needle syringe permanently installed in the septum (to avoid loosing BVOCs when exchanging the $CO_2$-trap). 1 mL was transferred to each of two 2 mL Eppendorf tubes with either 0.7 mL water or 0.7

mL $BaCl_2$ (1.5M, to precipitate trapped $^{14}CO_2$ as $Ba^{14}CO_3$) to differentiate between trapped $^{14}CO_2$ and dissolved $^{14}$-BVOC (Fig. 2). After 5 h reaction time, the tubes were centrifuged (12000 g, 2 min) and 1 mL from each tube was counted by liquid scintillation. 1 mL 1M NaOH had been added to the scintillation liquid in the case of the tube added only water. This was done to increase pH in the liquid and thereby avoid losses of $^{14}CO_2$.



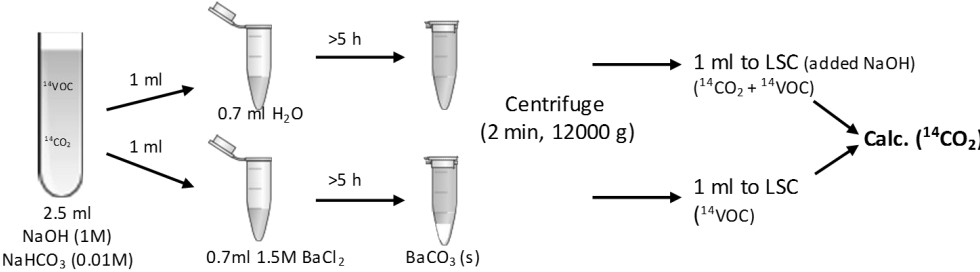


**Figure 2.** Sketch of method for capturing $^{14}CO_2$, separating it from dissolved $^{14}$C-BVOC and analyzing by liquid scintillation counting (LSC).

After the last sampling point, 30 mL methanol were added to the soil in each flask through the permanently installed
needle to extract any residual $^{14}$C-BVOC. After 24 h shaking the supernatant was transferred to a 50 mL centrifuge tube and centrifuged (4000 g, 5 min). The $^{14}$C-activity was then determined in 3 mL supernatant by liquid scintillation counting.

Incubations were made in three replicates. In addition to each BVOC/soil combination, a negative control was included
in which the soil had been sterilized by autoclaving twice. Oxygen consumption during incubation was determined for each soil type by incubating an additional flask in which oxygen spot sensors (PreSens, Regensburg, Germany) readable through the glass of the bottles, had been installed.

The incubation method is a modification of previous methods for measuring mineralization of organic compounds.
Suitability and limitations of the method are discussed in the supplementary information.

## 3 Results and Discussion

### 3.1 Soil characterization

The four top-soils used in the study showed clear differences with regards to major soil parameters like soil organic
matter content, pH and microbial biomass (Table 1). The two forest top soils differed in soil organic matter content, but were both acidic (pH just below 4). The Arctic Heath top soil had an organic matter content in between the two forest soils but a higher pH of 5.3. As expected, the Arctic bare soil differed most, as it comprised the parent mineral soil while the others were dominated by organic matter accumulation on top of the parent soil. Nevertheless, the bare soil did contain 3.6% soil organic matter and some bacterial biomass (Table 1), which may be due to its close proximity
(meter-scale) to vegetated areas. However, despite its relatively high content of organic matter and bacteria, it showed very low oxygen consumption during incubation (0.3 µM g$^{-1}$ dry weight (d.w.) d$^{-1}$) compared to the three organic soils (5-14 µM g$^{-1}$ d.w. d$^{-1}$). This indicates that the soil organic matter is not very reactive and/or that the bacterial activity





per cell is low in this soil. It should be stressed, that the measured oxygen consumption is not necessarily the same that it would be in nature, as the soil was disturbed (homogenized by hand) which may increase bioavailability of soil

organic matter. Fungal biomass (determined as ITS2 gene copies) was much higher in the coniferous soil compared to the other organic soil types, which may be expected as fungi are known to play a key role in degradation of needle litter (Boberg, 2009).

**Table 1.** Soil parameters determined from homogenized samples with major roots removed. Average of three replicate extractions
and analyses for 16s and ITS2 (±standard deviation) and one for the other parameters. Sample depth corresponds roughly to the depth of the organic layer after removing litter from the top and is the average depth of 10 pooled soil cores. At the Arctic bare soil no organic layer was present. $O_2$-consumption is measured during mineralization experiments and may be different from that in nature. All parameters except moisture are on dry weight basis.

| Soil | Depth (cm) | SOM (%) | $pH_{H2O}$ | Moisture (weight %) | 16s (copies g$^{-1}$) | ITS2 (copies g$^{-1}$) | $O_2$-consumption ($\mu M$ g$^{-1}$ d$^{-1}$) |
|---|---|---|---|---|---|---|---|
| Temp. conif. | 0-6 | 78 | 3.8 | 45 | $6.2 \cdot 10^{10} \pm 2.8 \cdot 10^{10}$ | $5.5 \cdot 10^{8} \pm 6.9 \cdot 10^{7}$ | 9 |
| Temp. beech | 0-5 | 20 | 3.9 | 46 | $4.2 \cdot 10^{10} \pm 1.2 \cdot 10^{10}$ | $3.7 \cdot 10^{7} \pm 1.7 \cdot 10^{7}$ | 5 |
| Arctic heath | 0-6 | 36 | 5.3 | 51 | $2.8 \cdot 10^{10} \pm 4.5 \cdot 10^{10}$ | $7.0 \cdot 10^{7} \pm 1.5 \cdot 10^{6}$ | 14 |
| Arctic bare | 0-5 | 3.6 | 7.3 | 8.1 | $2.2 \cdot 10^{9} \pm 1.1 \cdot 10^{9}$ | $2.4 \cdot 10^{6} \pm 6.8 \cdot 10^{5}$ | 0.3 |

**3.2 BVOC mineralization**

As model compounds we chose six BVOCs that have well described natural sources, are commonly detected in nature and have quite different molecular weights and physical/chemical properties (Table 2).





**Table 2.** Characteristics of the model compounds sorted by boiling point (BP). X means clear evidence for specified source in the environment. (X) means that some evidence exists. ∞ means unlimited solubility (miscible).

| Name | Cas. No. | BP (°C) | Molecular weight | $S_w$ (mg L$^{-1}$) | Plant source | Soil/microbial source | Anthropo-genic source |
|---|---|---|---|---|---|---|---|
| Chloroform | 67-66-3 | 61 | 119 | 8000 | (X)[k] | X[i,j] | X |
| Methanol | 67-56-1 | 65 | 32 | ∞ | X[f] | X[a,b,e] | X |
| p-cymene | 99-87-6 | 177 | 134 | 23 | X[g,m] | (X)[a]* | |
| Benzaldehyde | 100-52-7 | 178 | 106 | 3000 | X[f,m] | X[c,d] | X |
| Acetophenone | 98-86-2 | 202 | 120 | 5500 | X[m] | X[c,d] | X |
| Geraniol | 106-24-1 | 230 | 154 | 686 | X[f,h] | (X)[l]* | |

[a]Asensio et al., 2007. [b]Schink and Zeikus, 1980. [c]Gutiérrez-Luna et al., 2010. [d]McNeal and Herbert, 2009. [e]Bäck et al., 2010. [f]Kesselmeier and Staudt, 1999. [g]Ortega et al, 2008. [h]Chen and Viljoen, 2010. [i]Hoekstra et al., 1998. [j]Albers et al., 2010. [k]Laturnus and Matucha, 2008. [l]Schulz and Dickschat, 2007. [m]Jardine et al., 2010.

*Limited evidence for a soil or microbial source of these two monoterpenoids, but clear evidence for a general monoterpenoid production in soil and by various microorganisms (e.g. Schulz and Dickschat, 2007; Leff and Fierer, 2008; McNeal and Herbert, 2009; Bäck et al., 2010).

Five of the six BVOCs were rapidly mineralized in all four soils included in the mineralization experiment, with
chloroform showing somewhat slower mineralization (Fig. 3). None of the sterilized soil samples showed any detectable mineralization so the degradation of the BVOCs was in all cases microbially derived. However, great differences were observed with regards to speed of mineralization, extent of mineralization and variation between soil types. Methanol and benzaldehyde showed the highest mineralization rates. Especially for methanol (Fig. 3b), mineralization was so fast that the $CO_2$ transfer rate from soil to trap was most likely determining the shape of the
mineralization curve rather than the speed of mineralization. For example, the theoretical initial (0-2 h) mineralization rate in the Arctic Heath soil that can be determined with the applied method would be 40% h$^{-1}$ as calculated from the curve in Supplementary Fig. S1, and the observed mineralization of methanol in that soil type was 39% h$^{-1}$ (Table 3). Also benzaldehyde mineralization was so fast that probably the $CO_2$ transfer rate influenced the shape of the mineralization curve (Fig. 3d). The fact that methanol is degraded quickly in soil is not a surprise, as many isolated soil
bacteria have the capability to degrade this BVOC (Kolb, 2009), and different temperate grassland and forest soils have been found to contain at least 10$^6$ bacteria with the capability to degrade methanol per gram of soil (Stacheter et al., 2013). However, our data are the first to demonstrate degradation of methanol within the range of observed atmospheric concentrations (less than 100 ng L$^{-1}$, Seco et al. (2007)). Degradation of benzaldehyde in soil or by soil microorganisms has not been demonstrated, but benzaldehyde mineralization by pure microbial cultures has been shown (Kamada et al.,
220 2002).

Following methanol and benzaldehyde, geraniol and acetophenone had the highest mineralization rates with most of the mineralization occurring within the first 24 hours of incubation (Fig. 3e,f). These four rapidly degraded compounds had in common that no clear difference in mineralization rate was observed between the soil types (Table 3). For methanol
and partly for benzaldehyde this observation could be influenced by method limitations (too fast mineralization to be kept in pace by transfer of $CO_2$ to the trap may have masked any differences), however for geraniol and acetophenone



this was not the case. In other words, Arctic soils mineralized these compounds as quickly as temperate forest soils and perhaps even more interestingly, the Arctic bare soil showed similar mineralization rates as the organic soil types. This is despite a much lower abundance of microorganisms as determined by qPCR and a much lower microbial

heterotrophic activity during incubation as determined by oxygen consumption (Table 1). Geraniol mineralization has previously been investigated in soil sampled underneath *Populus tremula* tree crowns (Owen et al., 2007). The mineralization observed in that study was different from the one we observed, with an initial lag phase with less than 5% mineralization in the first ~10 hours. The lag phase was followed by maximum mineralization rates of 1-3% h$^{-1}$ which is close to what we observed right after the start of incubation (Table 3). An extremely high geraniol

concentration of 600 mg kg$^{-1}$ soil was used in that study compared to 6-11 µg kg$^{-1}$ soil in ours, which is the most likely cause of this difference in mineralization. The geraniol concentration used by Owen et al. (2007) would allow growth with geraniol as substrate (hence the lag phase) while the concentrations we used would allow only very limited microbial growth. However, the two studies all in all demonstrate that oxygenated monoterpenes may be degraded within a very large concentration range in soil.


P-cymene mineralization showed as the only BVOC clear differences between the soil types (Fig. 3c). Initial mineralization rates were by far the highest in the coniferous forest soil (10% h$^{-1}$, Table 3) followed by the Arctic heath soil (2% h$^{-1}$), the beech forest soil (0.4% h$^{-1}$) and the Arctic bare soil (0.2% h$^{-1}$). In other words, the coniferous forest soil showed a 25 times higher initial mineralization rate compared to the beech forest soil sampled just 750 meters

away. In addition, the three soils with slowest mineralization showed a slightly s-shaped mineralization curve meaning that mineralization rate increased after an initial lag-phase with slower mineralization (Fig. 3c and Table 3). All in all it appears that the coniferous forest soil is especially adapted to degrade p-cymene. P-cymene is a hydrocarbon monoterpene (monoterpene without heteroatoms) and these are emitted in very high concentrations in coniferous forests (Guenther et al., 1994; Rinne et al., 2009). Our measurements also showed a much higher concentration of this BVOC

group in the atmosphere of the coniferous forest compared to the other sampling sites (Table 4). In addition, needle litter emits high amounts of hydrocarbon monoterpenes (Aaltonen et al., 2011, Faiola et al., 2014) exposing the soil to these compounds found in higher concentrations in soil under conifers than deciduous trees (Smolander et al., 2006). All in all it seems likely that the high adaptation for p-cymene mineralization in the coniferous forest soil is caused by a high natural input of hydrocarbon monoterpenes to this soil type.


Chloroform, which is a well-known pollutant but also a natural product in soil (Hoekstra et al., 1998; Albers et al., 2010; Johnsen et al., 2016), was mineralized in all four soils (Fig. 3a), but at much slower rates compared to the other BVOCs (Table 3). Chloroform mineralization was previously determined at 10°C in a spruce forest soil in which initial mineralization rates of 0.01-0.04 % h$^{-1}$ were observed (Albers et al., 2011). These rates are roughly ten times lower than

the ones observed in our study (0.2-0.5% h$^{-1}$, Table 3). The spruce forest soil was similar to the coniferous forest soil used in our study, but there was a difference in chloroform concentration, which in our case was 4-7 µg kg$^{-1}$ soil and in the previous study was 350 µg kg$^{-1}$ soil. This stresses that the concentration used during incubation may to a high degree determine how fast the compound is mineralized. On the other hand, if mineralization rates are recalculated to a mass-unit per time-unit, differences in the case of chloroform mineralization would be much smaller between the two

studies.



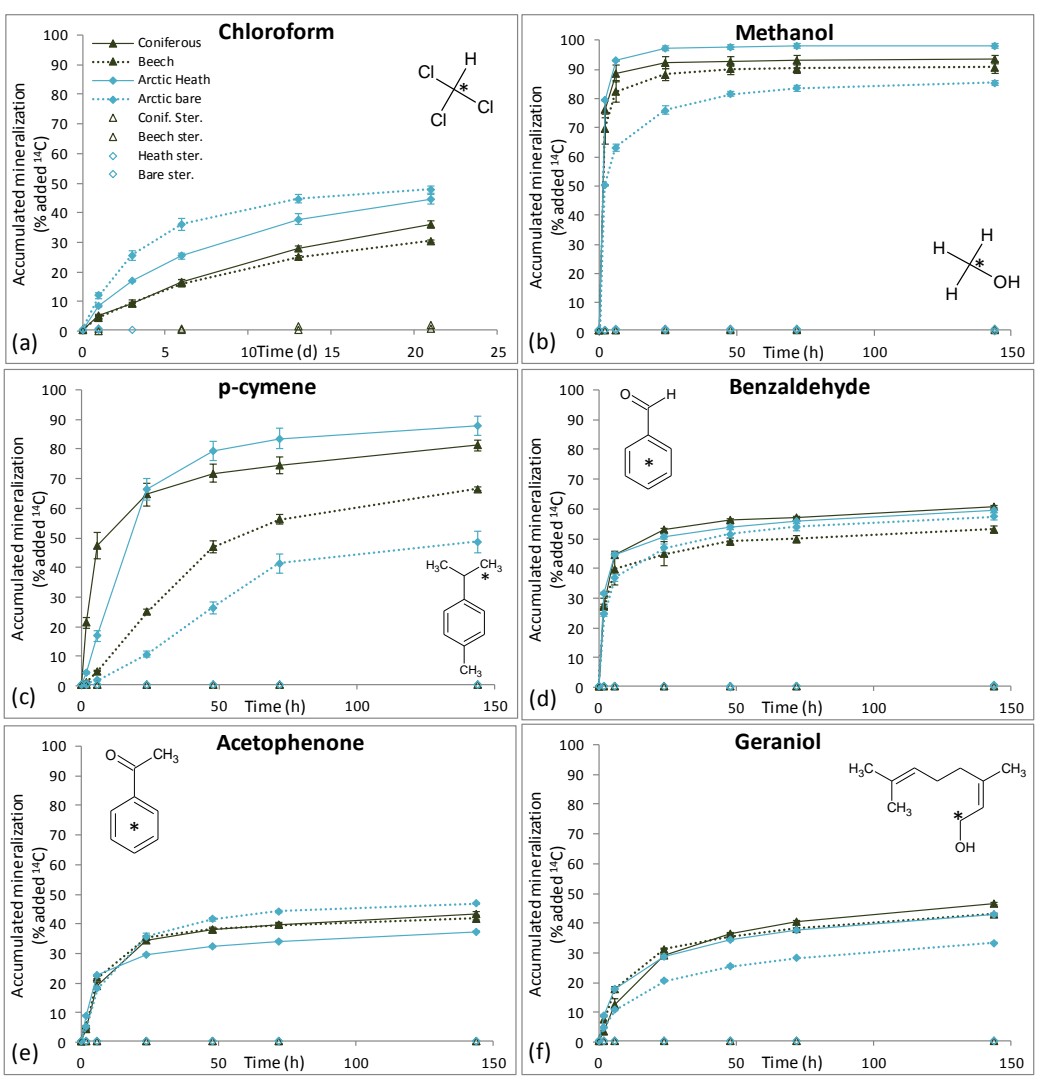

**Figure 3.** Mineralization curves for six BVOCs in soil from coniferous forest, beech forest, Arctic heath and Arctic bare soil. Initial BVOC concentrations varied from 0.8-11 µg kg soil$^{-1}$. "Ster." means that soil was sterilized twice by autoclaving. Error bars are standard deviation of triplicate incubations. Some error bars are smaller than the symbols.

The extent of mineralization (determined as $^{14}CO_2$ release at the termination of experiment) was in general similar between soils but differed greatly between the BVOCs. Methanol showed almost 100% $^{14}CO_2$ release while acetophenone and geraniol released only 40%. The 40% release should not be interpreted as if only 40% of the compound was degraded, but rather that 60% of the mineralized compound was used as a carbon source for microbial growth. This is a generally accepted interpretation of mineralization curves that often go to yields of only 40-50% with the remaining part incorporated into biomass that is only slowly mineralized along with microbial turnover (Nowak et





al., 2011; Glanville et al., 2016). Just after incubation, we extracted non-degraded or metabolized BVOCs with methanol, and only sterilized samples released a major pool of methanol-extractable $^{14}$C (typically between 50 and

95%) while non-sterilized samples released less than 5%. This strongly supports the interpretation that all BVOC was degraded. Geraniol was an exception from this with 15-25% of added $^{14}$C extracted in non-sterilized samples and 83-92% extracted in sterilized samples. While methanol, benzaldehyde, acetophenone and p-cymene were conclusively degraded completely within 140 hours (and presumably much faster) we therefore cannot exclude the possibility that some less degradable degradation products of geraniol have accumulated. It has been shown that some fungi have the

ability to metabolize geraniol into various derivatives (Demyttenaere et al., 2000).

Based on extent of mineralization, some compounds (e.g. methanol) were used only as a source of energy (as electron donor), while others (e.g. geraniol and acetophenone) were also used as a carbon source for growth. Recently, Gunina et al. (2017) suggested that the oxidation state of a C-atom determines how much is released as $CO_2$, and how much is

incorporated into biomass. They found a positive relationship between carbon oxidation state and $^{14}CO_2$-release for seven easily degradable low molecular weight sugars, acids and amino acids. However, the carbon atom in methanol (oxidation state -2) is more reduced than the labeled carbon atoms in geraniol (oxidation state 0), benzaldehyde and acetophenone (both -1), so the oxidation state does not determine mineralization extent of the model BVOCs.

P-cymene was an exception from the minor difference in mineralization extent between soil types. In soil from the coniferous forest and from the Arctic heath, more than 80% of the $^{14}$C was liberated as $^{14}CO_2$ (Fig. 3c). In the Arctic bare soil only half of this release was measured, while the beech forest soil was in between. In all soils, p-cymene dissipation was complete at the end of the experiment. One possible explanation for this difference is that different microorganisms degrade p-cymene in the studied soils and that these different organisms have different degradation

strategies for the compound, i.e. different fractions used for energy and growth. Another, perhaps more likely explanation, is that p-cymene is used as a carbon source mainly when degradation is occurring along with microbial growth. This explanation is supported by the fact that the slower the initial degradation is and the more s-shaped the mineralization curves are (presence of lag phase, Table 3), the more carbon seems to be accumulated into biomass (less $^{14}CO_2$-release, Fig. 3c). This is also supported by the earlier observed higher mineralization extent of $^{14}$C- geraniol at

high concentration that supported growth (mineralization extent of 64-75%, Owen et al., 2007) compared to the mineralization extent we observed for this compound with no or very little growth (33-46%, Fig. 3f). In addition, the highest mineralization extent in the case of geraniol was observed in the coniferous forest soil, which was the only soil where a lag phase (though very weak) was observed (Table 3).





**Table 3.** Mineralization parameters calculated from the mineralization experiment shown in Fig. 3. Initial mineralization rate is calculated as the average rate during the first two hours of incubation. A lag phase is noted where the initial mineralization rate is not the highest. (Yes) denotes a very weak lag phase.

|  | Initial mineralization rate (% $h^{-1}$) | | | | Lag phase? | | | |
|  | Conif. | Beech | Heath | Bare | Conif. | Beech | Heath | Bare |
| --- | --- | --- | --- | --- | --- | --- | --- | --- |
| Chloroform | 0.20 | 0.17 | 0.34 | 0.48 | No | No | No | No |
| Methanol | 38 | 35 | 39 | 25 | No | No | No | No |
| p-Cymene | 10 | 0.4 | 2.0 | 0.2 | No | Yes | (Yes) | Yes |
| Benzaldehyde | 14 | 13 | 16 | 12 | No | No | No | No |
| Acetophenone | 2.0 | 2.6 | 4.3 | 2.3 | (Yes) | (Yes) | No | (Yes) |
| Geraniol | 1.5 | 3.5 | 4.3 | 2.3 | (Yes) | No | No | No |

The potential for very fast mineralization of different BVOCs in different temperate and Arctic soils may have

significant environmental implications. A few previous studies have shown deposition of BVOCs onto soil (Ramirez et al., 2010; Spielmann et al., 2017) or a bidirectional atmosphere/soil exchange of certain BVOCs (Asensio et al., 2007; Asensio et al., 2008; Gray et al., 2014), but the mechanism behind the uptake of BVOCs into or onto soil has been largely uninvestigated. Our results suggest that BVOCs will be taken up from the atmosphere by microorganisms that then mineralize the compounds. The concentration of BVOCs in the atmosphere is very low, also at the sites where we

sampled soil (Table 4). Mineralization experiments cannot be carried out at such low concentrations but we used BVOC concentrations that are much more realistic than those used in previous degradation studies. Furthermore, similar atmospheric concentrations as we used for incubations have been observed in nature for methanol (Seco et al., 2007), chloroform (Albers et al., 2011) and monoterpenes (Barney et al., 2009).

It is therefore very likely that soil microorganisms also take up and mineralize BVOCs in the natural environment and most likely also in urban environments, where concentrations in the air can be much higher due to additional anthropogenic input (Seco et al., 2007). *In situ* uptake studies using e.g. PTR-MS technology should be carried out in order to provide quantitative estimates of the importance of BVOC uptake in soil. However, simultaneous formation and degradation of the compounds is a complicating aspect in such studies. The use of labeled compounds in the field to

determine simultaneous formation and degradation, as previously done in laboratory studies with methane (von Fischer and Hedin, 2002) and methyl halides (Rhew et al., 2003), could be a great supplement to more conventional PTR-MS studies.





**Table 4.** Atmospheric concentrations of relevant BVOCs (mean ± standard deviation, n=3) measured 10 cm above soil surface (coniferous, beech and Arctic bare sites) or 5 cm above the canopy (Arctic heath site) the day of soil sampling. Methanol could not be analyzed with the applied methods.

| Name | Atmospheric concentration (ng L$^{-1}$) | | |
| --- | --- | --- | --- |
| | Coniferous* | Beech | Arctic** |
| Oxygenated monoterpenes | 0.00 ±0.00 | 0.00 ±0.00 | 0.01 ±0.01 |
| Hydrocarbon monoterpenes | 3.36[a] ±0.32 | 0.37[b] ±0.12 | 0.71[c] ±0.10 |
| Benzaldehyde | 1.01 ±0.03 | 1.14 ±0.08 | 0.00 ±0.00 |
| Acetophenone | 0.44 ±0.06 | 0.59 ±0.03 | 0.01 ±0.01 |
| Chloroform | 0.10 ±0.02 | 0.06 ±0.00 | 0.06 ±0.00 |

*n=2 due to loss of a sample, except for chloroform (n=3). **One sample from the bare soil, two from the Arctic Heath. [a]Mainly pinenes, camphene, carene and p-cymene. [b]Mainly camphene, α-pinene, δ-terpinene and carene. [c]Mainly δ-terpinene.

In addition to the uptake from the atmosphere, the very fast mineralization rates are likely important in shaping the net BVOC emissions from soil. The net BVOC release from soil to the atmosphere in general is low compared to the plant emissions (Peñuelas et al., 2014), but emissions may represent a minor portion of the amount that was excreted by soil microbes (Insam and Seewald, 2010; Garbeva et al., 2014) or by roots (Lin et al., 2007; Delory et al., 2016), produced for example with the purpose of communication. It is thus possible that BVOCs are a significant source of carbon to soil microbes and hence that BVOC formation and degradation may be an important but little recognized part of internal carbon cycling in soil. In addition, plant litter releases BVOCs from both abiotic and biotic processes (for example terpenoids (Faiola et al., 2013) and methanol (Gray et al., 2010)). These BVOCs may to a large degree never reach the atmosphere but rather be an input of degradable carbon to microorganisms in the top soil.

**4 Conclusions**

In conclusion, we have shown that six chemically very different BVOCs are all mineralized by microbes in Arctic and temperate soils at environmentally relevant concentrations. Five of the BVOCs were mineralized very quickly, but still we observed a relatively large compound-to-compound variation in mineralization rate as well as mineralization extent compared to a much lower soil-to-soil variation. P-cymene was an exception from this pattern with both mineralization rate and extent differing between soils of different origin.

It is thus clear that soil microbes are able to degrade completely and quickly BVOCs released by aboveground vegetation, soil microbes and plant roots and additional studies should be carried out to quantify this process in nature. In addition to the possible atmospheric implications of BVOC degradation by soil microbes, BVOC formation and degradation may be an important but little recognized part of internal carbon cycling in soil.



*Acknowledgments.* We wish to thank Pia Bach Jacobsen, GEUS, for excellent technical assistance in the laboratory. We also thank Anders Priemé, Copenhagen University for advice on the qPCR targeting the fungal community and for providing the ITS2 standard. The study was financially supported by the Danish Council for Independent Research | Natural Sciences (DFF-4002-00495), the Villum foundation (VKR022589), and by the Danish National Research

Foundation (CENPERM DNRF100).

*Competing interests.* The authors declare that they have no conflict of interest.

*Data availability.* The data set related to Figure 3 has been provided as a supplement.

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
