# Peer review of "Rapid mineralization of biogenic volatile organic compounds in temperate and Arctic soils"

_Biogeosciences, 2018_

## Referee Comment (RC1) · Anonymous Referee #1 · 18 Mar 2018

General comments

This paper investigates the mineralization of five BVOCs in four different soils. By using BVOCs labelled with 14C the paper clearly shows the mineralization process of BVOCs occurring in the different soil types. The methodology is adequately described and the experimental procedures are well written, as well as the introduction, results and discussion sections. The fact that soil microbes can metabolize soil BVOCs is already known but, as the authors say, there are still not many studies directly proving microbial degradation of BVOCs. The authors say that one important value of this study is that the incubated soils were exposed to realistic environmental concentrations of BVOCs, not like the other studies, where higher concentrations were used. It would be interesting for the readers to have a table were one could see the real initial

concentration (not a range), together with the concentration measured of these BVOCs in the environment (in this experiment or in the literature, if some atmospheric BVOC measurement is missing), the amount of BVOC metabolized to CO2 and the amount of BVOC extracted at the end of the incubation, for each BVOCs and soil incubated. This information would help to evaluate the main points of the paper: that some BVOCs can be degraded completely in soil (by giving the recovered BVOCs at the end of the experiment one could see how much was in incorporated in the microbial biomass or how much was adsorbed to soil particles) at the relevant environmental concentrations measured. Regarding this point, for the BVOCs that were measured in the atmosphere (not methanol), the highest atmospheric concentrations shown in Table 4 (around 3 ng/L in the coniferous forest, measured at 10 cm above soil) would be still 21 times lower than the minimum concentration used in the incubations (64 ng/L). There are of course technical difficulties, as the authors say, to measure mineralization at the very low atmospheric concentration, thus that table would highlight to what extent the authors have narrowed this challenge.

Specific comments

Page 4, line 115: Table 1 instead of Table 2

Page 5, line 137: As suggested in the general comments, it would be nice to have a table with the initial concentrations for each BVOC and soil type, the corresponding atmospheric concentrations, etc.

Page 5, line 138: This range 0.8-11 ug/Kg soil FW, is environmentally realistic? Regarding monoterpenes White (1991) reports 12-47 ug/g soil DW in the organic horizon and 0.03 to 0.23 ug/g mineral horizon of Ponderosa pine forest. That would be much higher than the values in this experiment, at least for the organic layer. Is there any information for other BVOCs ? But I guess it's very difficult to find this information... Reference: White, C.S., 1991. The role of monoterpenes in soil-nitrogen cycling processes in ponderosa pine results from laboratory bioassays and field studies. Biogeo-

chemistry 12 (1), 43–68.

Page 6, line 170: Table 2 instead of Table 1

Page 8, line 208-213: It is very nice to read the investigations about the transfer rate of $CO_2$ in the supplementary information Fig. S1.

Page 11, line 280: This 5% is then adsorption to soil? Again, a table to compare values between soils and BVOCs would be useful in my opinion

Page 11, line 298: what do you mean by dissipation? Do you mean you recovered all the p-cymene added with the extraction?

Page 13, Table 4: Assuming the 3 columns represent the sites where BVOCs were measured at 10 cm surface, where is the Arctic heath site?
* * *

---

## Author Comment (AC1) · 11 Apr 2018

We have included the Referee comments in the reply below. Each comment is followed by a comment by us and a desription of which actions we will take when revising the manuscript. The reply is also attached as a separate pdf-file where the new Tables are included as well.

General comments This paper investigates the mineralization of five BVOCs in four different soils. By using BVOCs labelled with 14C the paper clearly shows the mineralization process of BVOCs occurring in the different soil types. The methodology is adequately described and the experimental procedures are well written, as well as the introduction, results and discussion sections. The fact that soil microbes can metabo-

lize soil BVOCs is already known but, as the authors say, there are still not many studies directly proving microbial degradation of BVOCs. The authors say that one important value of this study is that the incubated soils were exposed to realistic environmental concentrations of BVOCs, not like the other studies, where higher concentrations were used. It would be interesting for the readers to have a table were one could see the real initial concentration (not a range), together with the concentration measured of these BVOCs in the environment (in this experiment or in the literature, if some atmospheric BVOC measurement is missing), the amount of BVOC metabolized to CO2 and the amount of BVOC extracted at the end of the incubation, for each BVOCs and soil incubated. This information would help to evaluate the main points of the paper: that some BVOCs can be degraded completely in soil (by giving the recovered BVOCs at the end of the experiment one could see how much was in incorporated in the microbial biomass or how much was adsorbed to soil particles) at the relevant environmental concentrations measured. Regarding this point, for the BVOCs that were measured in the atmosphere (not methanol), the highest atmospheric concentrations shown in Table 4 (around 3 ng/L in the coniferous forest, measured at 10 cm above soil) would be still 21 times lower than the minimum concentration used in the incubations (64 ng/L). There are of course technical difficulties, as the authors say, to measure mineralization at the very low atmospheric concentration, thus that table would highlight to what extent the authors have narrowed this challenge.

Our reply to the general comments: We have considered carefully to make an additional table with the data suggested by the referee. We are afraid that such a table will increase confusion rather than make things clearer, since very different things would be compared in the table (environmental BVOC-concentrations, BVOC concentrations during incubation, mineralized fraction of BVOC and extractable 14C). We do understand the desire to compare environmental and experimental concentrations. However, this is a difficult task, since we do not know how much of the BVOC we add that is in the air phase as we already discuss in the manuscript (see also our response to the specific comment concerning "page 5, line 138"). After all, we have decided to compare the two concentration types in Table 4, where environmental concentrations were already shown. We have also included comparable literature data, if available.

The mineralization extent is clearly an important parameter, but since it is relatively easy to read out of the curves in Figure 3, we did not include them in a table. We can see the need to be able to find the exact data, however and also the benefit of comparing with extraction data, although these play a minor role in the manuscript. We will therefore include a supplementary table with these data in the revised manuscript.

Specific comments Page 4, line 115: Table 1 instead of Table 2 Reply: The reviewer is correct, Table 1 and 2 should exchange numbers.

Page 5, line 137: As suggested in the general comments, it would be nice to have a table with the initial concentrations for each BVOC and soil type, the corresponding atmospheric concentrations, etc. Reply: These are now gathered in Table 4 (see response to "General comments")

Page 5, line 138: This range 0.8-11 ug/Kg soil FW, is environmentally realistic? Regarding monoterpenes White (1991) reports 12-47 ug/g soil DW in the organic horizon and 0.03 to 0.23 ug/g mineral horizon of Ponderosa pine forest. That would be much higher than the values in this experiment, at least for the organic layer. Is there any information for other BVOCs ? But I guess it's very difficult to find this information. . . Reference: White, C.S., 1991. The role of monoterpenes in soil-nitrogen cycling processes in ponderosa pine results from laboratory bioassays and field studies. Biogeochemistry 12 (1), 43–68.

Reply: This point raised by the reviewer is relevant and something we have given a great deal of thought when designing the experiment. First of all, BVOC measurements around or below the soil surface are very scarce in the literature and for the few studies that exist about BVOC concentrations in soil, very different methods have been used, making it impossible to compare. For example in the reference mentioned by the referee, BVOCs were extracted with an organic solvent from the solid matrix after

homogenization, hence representing only sorbed/bound BVOC as well as compounds in root material if any remained in the soil, while others measured the concentration in soil air, hence excluding sorbed/bound BVOC. Our focus in the study was the possible uptake from the atmosphere rather than degradation of sorbed/bound residues, although in nature it will all be connected, of course. Based on atmospheric concentration measurements, the concentration of many BVOCs seems in general lower than what is possible to analyze with 14C-mineralization experiments. We therefore used the lowest concentration possible, which for some of the compounds is not far from realistic and under all circumstances much more realistic than what has been used in previous degradation studies. Furthermore, a fraction of the BVOC we add will be sorbed and/or dissolved in soil water, hence making actual atmospheric concentrations in the experiment even closer to those in nature. However, the exact concentration in atmosphere and soil during incubation is not easy to calculate and furthermore it will change rapidly due to the rapid degradation.

In the manuscript, we already state the following, which we believe sums up these considerations (lines 136-139 and 319-323): "The BVOC concentrations used for incubation corresponded to 43-73 ppbv (64-504 ng L-1), assuming all BVOC was present in the headspace of the flasks. Most of the BVOCs were, however likely dissolved in water or adsorbed to the soil, so recalculating to a soil basis (0.8-11 $\mu$g kg-1 f.w. soil) may be more appropriate." And: "The concentration of BVOCs in the atmosphere is very low, also at the sites where we sampled soil (Table 4). Mineralization experiments cannot be carried out at such low concentrations but we used BVOC concentrations that are much more realistic than those used in previous degradation studies. Furthermore, similar atmospheric concentrations as we used for incubations have been observed in nature for methanol (Seco et al., 2007), chloroform (Albers et al., 2011) and monoterpenes (Barney et al., 2009)."

Page 6, line 170: Table 2 instead of Table 1 Reply: The reviewer is correct, Table 1 and 2 should exchange numbers.

Page 8, line 208-213: It is very nice to read the investigations about the transfer rate of CO2 in the supplementary information Fig. S1. Reply: Thank you. We did consider to include it in the manuscript itself, but ended up placing it in the supplementary, where the interested reader will hopefully find it. . .

Page 11, line 280: This 5% is then adsorption to soil? Again, a table to compare values between soils and BVOCs would be useful in my opinion Reply: These 5% could be sorbed/bound mother compound or (more likely) microbial metabolites, containing 14C due to utilization of the 14C-BVOC. Regarding the Table, we have made a supplementary table (see response to "General comments"). We will add the following sentence: "These 5% could be sorbed/bound mother compound or (more likely) microbial metabolites, containing 14C due to utilization of the 14C-BVOC."

Page 11, line 298: what do you mean by dissipation? Do you mean you recovered all the p-cymene added with the extraction? Reply: By dissipation we mean degradation in the sense of disappearance, or in other words that we could not recover the compound by extraction. The word "degradation" would probably be easier to read, but since we only measure that the compound is gone, we used the word "dissipation". To clarify, we will change the sentence as follows: "In all soils, p-cymene dissipation was complete at the end of the experiment," Will be changed to: "In all soils, p-cymene degradation was complete at the end of the experiment, since we could not extract any 14C with methanol."

Page 13, Table 4: Assuming the 3 columns represent the sites where BVOCs were measured at 10 cm surface, where is the Arctic heath site? Reply: As explained in the footnotes of the Table, this is an average of one sample from the bare soil and two from the Arctic Heath. We did take more samples, but they were unfortunately destroyed during transport, and since there was no major difference in compound composition and concentration between the three samples (which may also be expected since there is no forest canopy and the sites are located just 300 m apart) we decided to treat it as one sample type.

Please also note the supplement to this comment:
https://www.biogeosciences-discuss.net/bg-2018-32/bg-2018-32-AC1-supplement.pdf
* * *
[Figure]

**Supplement:**

Reply to referee 1

**General comments**

This paper investigates the mineralization of five BVOCs in four different soils. By using BVOCs labelled with 14C the paper clearly shows the mineralization process of BVOCs occurring in the different soil types. The methodology is adequately described and the experimental procedures are well written, as well as the introduction, results and discussion sections. The fact that soil microbes can metabolize soil BVOCs is already known but, as the authors say, there are still not many studies directly proving microbial degradation of BVOCs. The authors say that one important value of this study is that the incubated soils were exposed to realistic environmental concentrations of BVOCs, not like the other studies, where higher concentrations were used. It would be interesting for the readers to have a table were one could see the real initial concentration (not a range), together with the concentration measured of these BVOCs in the environment (in this experiment or in the literature, if some atmospheric BVOC measurement is missing), the amount of BVOC metabolized to CO2 and the amount of BVOC extracted at the end of the incubation, for each BVOCs and soil incubated. This information would help to evaluate the main points of the paper: that some BVOCs can be degraded completely in soil (by giving the recovered BVOCs at the end of the experiment one could see how much was in incorporated in the microbial biomass or how much was adsorbed to soil particles) at the relevant environmental concentrations measured. Regarding this point, for the BVOCs that were measured in the atmosphere (not methanol), the highest atmospheric concentrations shown in Table 4 (around 3 ng/L in the coniferous forest, measured at 10 cm above soil) would be still 21 times lower than the minimum concentration used in the incubations (64 ng/L). There are of course technical difficulties, as the authors say, to measure mineralization at the very low atmospheric concentration, thus that table would highlight to what extent the authors have narrowed this challenge.

We have considered carefully to make an additional table with the data suggested by the referee. We are afraid that such a table will increase confusion rather than make things clearer, since very different things would be compared in the table (environmental BVOC-concentrations, BVOC concentrations during incubation, mineralized fraction of BVOC and extractable $^{14}$C). We do understand the desire to compare environmental and experimental concentrations. However, this is a difficult task, since we do not know how much of the BVOC we add that is in the air phase as we already discuss in the manuscript (see also our response to the specific comment concerning "page 5, line 138"). After all, we have decided to compare the two concentration types in Table 4, where environmental concentrations were already shown. We have also included comparable literature data, if available:

**Table 4.** Atmospheric concentrations of relevant BVOCs (mean ± standard deviation, n=3) measured 10 cm above soil surface (coniferous, beech and Arctic bare sites) or 5 cm above the canopy (Arctic heath site) the day of soil sampling. Methanol could not be analyzed with the applied methods. Comparable literature data are included, when available.

| Name | Atmospheric concentration (ng L$^{-1}$) | | | Initial headspace concentration during incubation (ng L$^{-1}$)*** |
|---|---|---|---|---|
| | Coniferous* | Beech | Arctic** | |
| Oxygenated monoterpenes | 0.00[e] ±0.00 | 0.00 ±0.00 | 0.01 ±0.01 | 504 |
| Hydrocarbon monoterpenes[d] | 3.36[a,f] ±0.32 | 0.37[b] ±0.12 | 0.71[c] ±0.10 | 260 |
| Benzaldehyde | 1.01 ±0.03 | 1.14 ±0.08 | 0.00 ±0.00 | 286 |
| Acetophenone | 0.44 ±0.06 | 0.59 ±0.03 | 0.01 ±0.01 | 350 |
| Chloroform | 0.10[g] ±0.02 | 0.06 ±0.00 | 0.06 ±0.00 | 340 |
| Methanol (literature data) | 0.3-284 (Seco et al., 2007) | | | 64 |

*n=2 due to loss of a sample, except for chloroform (n=3). **One sample from the bare soil, two from the Arctic Heath. ***Assuming all added BVOC is present in headspace, although most will likely be adsorbed to soil or dissolved in water. [a]Mainly pinenes, camphene, carene and p-cymene. [b]Mainly camphene, α-pinene, δ-terpinene and carene. [c]Mainly δ-terpinene. [d]Comparable literature values but from a different ecosystem type go from 0.5-50 ng L$^{-1}$ (Barney et al., 2009). [e]Air samples taken at the interface between litter and atmosphere have shown concentrations of 60-390 ng L$^{-1}$ (Ketola et al., 2011). [f]Air samples taken at the interface between litter and atmosphere have shown concentrations of 10-24300 ng L$^{-1}$ (Ketola et al., 2011). [g]Comparable literature data go from 0.08-2.1 ng L$^{-1}$ (Albers et al., 2010).

The mineralization extent is clearly an important parameter, but since it is relatively easy to read out of the curves in Figure 3, we did not include them in a table. We can see the need to be able to find the exact data, however and also the benefit of comparing with extraction data, although these play a minor role in the manuscript. We will therefore include a supplementary table with these data in the revised manuscript:

Table S1. Supplementary mineralization parameters from the mineralization experiment shown in Fig. 3. Mineralization extent is the accumulated liberation of $^{14}CO_2$ at the end of experiment (144 h, except chloroform (21 d)). Final extraction is the amount of $^{14}C$ extractable with methanol at the end of experiment. All data are average of three replicate incubations. nd = no data.

| | Mineralization extent (%) | | | | Extractable $^{14}C$ (%) | | | |
|---|---|---|---|---|---|---|---|---|
| | Conif. | Beech | Heath | Bare | Conif. | Beech | Heath | Bare |
| Chloroform | 36 | 30 | 44 | 47 | nd | nd | nd | nd |
| Methanol | 93 | 90 | 98 | 85 | 1 | 1 | 0 | 1 |
| p-Cymene | 81 | 66 | 87 | 48 | 8 | 7 | 4 | 5 |
| Benzaldehyde | 60 | 53 | 59 | 57 | 4 | 4 | 2 | 1 |
| Acetophenone | 43 | 41 | 37 | 46 | 6 | 4 | 3 | 4 |
| Geraniol | 46 | 42 | 42 | 33 | 19 | 17 | 16 | 25 |

**Specific comments**

Page 4, line 115: Table 1 instead of Table 2

Reply: The reviewer is correct, Table 1 and 2 should exchange numbers.

Page 5, line 137: As suggested in the general comments, it would be nice to have a table with the initial concentrations for each BVOC and soil type, the corresponding atmospheric concentrations, etc.

These are now gathered in table 4 (see response to "General comments")

Page 5, line 138: This range 0.8-11 ug/Kg soil FW, is environmentally realistic? Regarding monoterpenes White (1991) reports 12-47 ug/g soil DW in the organic horizon and 0.03 to 0.23 ug/g mineral horizon of Ponderosa pine forest. That would be much higher than the values in this experiment, at least for the organic layer. Is there any information for other BVOCs ? But I guess it's very difficult to find this information. . .

Reference: White, C.S., 1991. The role of monoterpenes in soil-nitrogen cycling processes in ponderosa pine results from laboratory bioassays and field studies. Biogeochemistry 12 (1), 43–68.

This point raised by the reviewer is relevant and something we have given a great deal of thought when designing the experiment. First of all, BVOC measurements around or below the soil surface are very scarce in the literature and for the few studies that exist about BVOC concentrations in soil, very different methods have been used, making it impossible to compare. For example in the reference mentioned by the referee, BVOCs were extracted with an organic solvent from the solid matrix after homogenization, hence representing only sorbed/bound BVOC as well as compounds in root material if any remained in the soil, while others measured the concentration in soil air, hence excluding sorbed/bound BVOC. Our focus in the study was the possible uptake from the atmosphere rather than degradation of sorbed/bound residues, although in nature it will all be connected, of course. Based on atmospheric concentration measurements, the concentration of many BVOCs seems in general lower than what is possible to analyze with $^{14}$C-mineralization experiments. We therefore used the lowest concentration possible, which for some of the compounds is not far from realistic and under all circumstances much more realistic than what has been used in previous degradation studies. Furthermore, a fraction of the BVOC we add will be sorbed and/or dissolved in soil water, hence making actual atmospheric concentrations in the experiment even closer to those in nature. However, the exact concentration in atmosphere and soil during incubation is not easy to calculate and furthermore it will change rapidly due to the rapid degradation.

In the manuscript, we already state the following, which we believe sums up these considerations (lines 136-139 and 319-323):

*"The BVOC concentrations used for incubation corresponded to 43-73 ppbv (64-504 ng L-1), assuming all BVOC was present in the headspace of the flasks. Most of the BVOCs were, however likely dissolved in water or adsorbed to the soil, so recalculating to a soil basis (0.8-11 µg kg-1 f.w. soil) may be more appropriate."*
And:
*"The concentration of BVOCs in the atmosphere is very low, also at the sites where we sampled soil (Table 4). Mineralization experiments cannot be carried out at such low concentrations but we used BVOC concentrations that are much more realistic than those used in previous degradation studies. Furthermore, similar atmospheric concentrations as we used for incubations have been observed in nature for methanol (Seco et al., 2007), chloroform (Albers et al., 2011) and monoterpenes (Barney et al., 2009)."*

Page 6, line 170: Table 2 instead of Table 1

Reply: The reviewer is correct, Table 1 and 2 should exchange numbers.

Page 8, line 208-213: It is very nice to read the investigations about the transfer rate of CO2 in the supplementary information Fig. S1.

Reply: Thank you. We did consider to include it in the manuscript itself, but ended up placing it in the supplementary, where the interested reader will hopefully find it…

Page 11, line 280: This 5% is then adsorption to soil? Again, a table to compare values between soils and BVOCs would be useful in my opinion

These 5% could be sorbed/bound mother compound or (more likely) microbial metabolites, containing $^{14}$C due to utilization of the $^{14}$C-BVOC. Regarding the Table, we have made a supplementary table (see response to "General comments").

We will add the following sentence:

"*These 5% could be sorbed/bound mother compound or (more likely) microbial metabolites, containing $^{14}$C due to utilization of the $^{14}$C-BVOC.*"

Page 11, line 298: what do you mean by dissipation? Do you mean you recovered all the p-cymene added with the extraction?

By dissipation we mean degradation in the sense of disappearance, or in other words that we could not recover the compound by extraction. The word "degradation" would probably be easier to read, but since we only measure that the compound is gone, we used the word "dissipation". To clarify, we will change the sentence as follows:

"*In all soils, p-cymene dissipation was complete at the end of the experiment,*"

Will be changed to:

"*In all soils, p-cymene degradation was complete at the end of the experiment, since we could not extract any $^{14}$C with methanol.*"

Page 13, Table 4: Assuming the 3 columns represent the sites where BVOCs were measured at 10 cm surface, where is the Arctic heath site?

As explained in the footnotes of the Table, this is an average of one sample from the bare soil and two from the Arctic Heath. We did take more samples, but they were unfortunately destroyed during transport, and since there was no major difference in compound composition and concentration between the three samples (which may also be expected since there is no forest canopy and the sites are located just 300 m apart) we decided to treat it as one sample type.

---

## Referee Comment (RC2) · M. Mäki (Referee) · 7 May 2018

General comments

The manuscript shows rapid mineralization of different BVOCs in temperate and Arctic soils. The manuscript is concise and clear. I appreciate your chosen scientific approach and use of relatively low BVOC concentrations, which are more realistic compared to the earlier studies. I recommend this manuscript for publication after it has been modified. Scientific significance of the manuscript would have been stronger if the number of BVOCs and soil types studied would be higher. Considering the Table 4, I would like you to justify, why you decided to choose the compounds that were not the dominating ones in the ambient air close to the soil surface. Why to choose p-cymene

if several other monoterpenes showed much higher concentrations in the atmosphere above the sampled soils? Especially when you say in the conclusions that BVOC degradation by soil microbes could have atmospheric implications. I would also like to read your reasoning behind why you decided to study only six different BVOCs when the spectrum of different BVOCs emitted by vegetation and soil processes is very high. One value of this study is that you studied different soil types. You should mention different soil types already in the abstract.

Specific comments

Line 12. You wrote in the text: "Their release into the atmosphere is important with regards to a number of physical and chemical processes." Please keep in mind that you will sell your manuscript to your readers. Please be more precise. What do you mean with this?

Line 33. Please remove "though".

Lines 37-38. Please clarify that this is a chain reaction from BVOCs and oxidants (OH, O3, NOx) to SOA and from there to cloud formation and properties.

Line 41. "Owe to" is not good. Please use another verb.

Line 43. Please clarify what is a fate model.

Lines 44-45. You wrote that "The microbial degradability of BVOCs - and especially the rate of degradation - are on the other hand very difficult to predict." Could you please clarify why microbial degradability of BVOCs is difficult to predict? In soil, there is a high diversity of compounds with varying properties for microbial degradation. Microbial population diversity is high. Chemical transformation from one compound to another happens also in soil. Soil conditions vary in time, which can affect degradability of BVOCs.

Lines 54-55. Field study or laboratory measurements? Which ecosystems/soil types? Please clarify.

Line 61. "The ultimate proof" is not scientific language.

Table 1. Please specify in the table that 16s is bacterial biomass and ITS2 is fungal biomass.

Line 105. "A snap-shot" is not scientific language.

Line 110. Please be more precise: a gas chromatograph–mass spectrometer, and please include the instrument details.

Line 137. Please correct "all BVOC was present". It should be: all the BVOCs were present in the headspace of the flasks.

Lines 210-220. It would be more easy to read if you would discuss methanol first and benzaldehyde after that. Now you discuss methanol first, then benzaldehyde, then methanol again and so on.

Line 344. You talk about communication between soil organisms. Please add a reference.

Line 327. PTR-MS should be the proton-transfer reaction mass-spectrometer.

Line 351: It is needless to say "In conclusion", when the title is Conclusions. Please make the conclusions more concise.

Table 2. Please specify in the table that Sw means water solubility.

Table 4. You could consider to add reactivity of each compound or reactivity range of each compound group, because it will likely affect your results. You should also present analytical methods and calculations in the M&M section.

Figure 3. You didn't do any statistical analysis on how the BVOCs behave in the different soil types. You should use valid statistical tests and add p values into the text. Please include statistical methods into the M&M section. Please remove the framing. Same for the Figure S1 in supplements. You should clarify in the figure caption that

chloroform was measured for 25 days and others for 150 hours.

Finally, it would be nice to see a map that shows locations of the sampling sites in supplements.

―――――――――――――――――――

---

## Author Comment (AC2) · 22 May 2018

Response is also attached as PDF-file for better overview.

Response to referee 2

M. Mäki (Referee) mari.maki@helsinki.fi General comments The manuscript shows rapid mineralization of different BVOCs in temperate and Arctic soils. The manuscript is concise and clear. I appreciate your chosen scientific approach and use of relatively low BVOC concentrations, which are more realistic compared to the earlier studies. I recommend this manuscript for publication after it has been modified. Scientific significance of the manuscript would have been stronger if the number of BVOCs and soil types studied would be higher. Considering the Table 4, I would like you to justify, why you decided to choose the compounds that were not the dominating ones in the ambient air close to the soil surface. Why to choose p-cymene if several other monoterpenes showed much higher concentrations in the atmosphere above the sampled soils? Especially when you say in the conclusions that BVOC degradation by soil microbes could have atmospheric implications. I would also like to read your reasoning behind why you decided to study only six different BVOCs when the spectrum of different BVOCs emitted by vegetation and soil processes is very high. One value of this study is that you studied different soil types. You should mention different soil types already in the abstract.

Our reply: With regards to the choice of model BVOCs we had the following major considerations. The first was the number of incubations we could handle in the laboratory within the manpower available to the study and six compounds in triplicate plus abiotic controls seemed like a good compromise. Then to choose these six compounds, we wanted some that represented major BVOC groups, since even though you can expect different degradation rates of compounds with similar chemical structures, you may after all expect larger differences between than within chemical groups. The last major consideration was then that the compound should be commercially available as 14C-labelled. The price of commercially available 14C-compounds is typically 1000-2000 Euros compared to non-available where you pay 10000-20000 Euro for a custom synthesis. The compounds we chose were those that may be considered most widespread / best group representatives and at the same time being commercially available. Even though it could have been nice to have a free choice of compound, this is simply not possible when working with 14C-labelled compounds unless you have a very large budget. . . With regards to the use of different soil types we state that in the abstract already.

Specific comments Line 12. You wrote in the text: "Their release into the atmosphere is important with regards to a number of physical and chemical processes." Please keep in mind that you will sell your manuscript to your readers. Please be more precise.

What do you mean with this?

Our reply: We understand this comment, however, in the Abstract, the general introduction should be limited, we believe. However "physical and chemical processes" Will be changed to: "climate related physical and chemical processes"

Line 33. Please remove "though". Our reply: Will be corrected as suggested

Lines 37-38. Please clarify that this is a chain reaction from BVOCs and oxidants (OH, O3, NOx) to SOA and from there to cloud formation and properties. Our reply: Will be corrected as suggested

Line 41. "Owe to" is not good. Please use another verb. Our reply: Will be corrected as suggested

Line 43. Please clarify what is a fate model. Our reply: "and fate models for these parameters could then be set up" will be deleted, as this part of the sentence is actually not necessary.

Lines 44-45. You wrote that "The microbial degradability of BVOCs - and especially the rate of degradation - are on the other hand very difficult to predict." Could you please clarify why microbial degradability of BVOCs is difficult to predict? In soil, there is a high diversity of compounds with varying properties for microbial degradation. Microbial population diversity is high. Chemical transformation from one compound to another happens also in soil. Soil conditions vary in time, which can affect degradability of BVOCs.

Our reply: The referee is correct that in most soils there is a huge potential for degradation of all sorts of organic compounds. Therefore it is also not a big surprise, if BVOCs are degraded in soil. However, the rate of degradation varies tremendously from compound to compound, and with the current QSAR models this is not predictable. Furthermore, the degradation rates may vary from soil to soil. This is what we mean by this sentence. We will add the following to the sentence to make this clearer: ", since

degradation rates in soil vary a lot from compound to compound and from soil to soil"

Lines 54-55. Field study or laboratory measurements? Which ecosystems/soil types? Please clarify. Our reply: Will be corrected as suggested

Line 61. "The ultimate proof" is not scientific language. Our reply: Will be changed as suggested

Table 1. Please specify in the table that 16s is bacterial biomass and ITS2 is fungal biomass. Our reply: Will be corrected as suggested

Line 105. "A snap-shot" is not scientific language. Our reply: We do not agree with the referee, this is often used and we can think of no better word.

Line 110. Please be more precise: a gas chromatograph–mass spectrometer, and please include the instrument details. Our reply: Information will be added as suggested.

Line 137. Please correct "all BVOC was present". It should be: all the BVOCs were present in the headspace of the flasks. Our reply: Will be corrected as suggested

Lines 210-220. It would be more easy to read if you would discuss methanol first and benzaldehyde after that. Now you discuss methanol first, then benzaldehyde, then methanol again and so on. Our reply: Will be corrected as suggested

Line 344. You talk about communication between soil organisms. Please add a reference. We will add two references, also included in the previous sentence (Garbeva et al., 2014; Delory et al., 2016).

Line 327. PTR-MS should be the proton-transfer reaction mass-spectrometer. Will be corrected as suggested

Line 351: It is needless to say "In conclusion", when the title is Conclusions. Please make the conclusions more concise. Will be corrected as suggested

Table 2. Please specify in the table that Sw means water solubility. Will be corrected as suggested

Table 4. You could consider to add reactivity of each compound or reactivity range of each compound group, because it will likely affect your results. You should also present analytical methods and calculations in the M&M section.

Our reply: We are not sure what the referee refers to. Is it the atmospheric reactivity of the compounds? If so, how could this affect our concentration measurements? We already present how the samples were taken and analysed in section 2.2 This section is now extended based on a previous comment by the referee. Since it is just concentration measurements (and not e.g. fluxes) there are no calculations involved.

Figure 3. You didn't do any statistical analysis on how the BVOCs behave in the different soil types. You should use valid statistical tests and add p values into the text. Please include statistical methods into the M&M section. Please remove the framing. Same for the Figure S1 in supplements. You should clarify in the figure caption that chloroform was measured for 25 days and others for 150 hours. Finally, it would be nice to see a map that shows locations of the sampling sites in supplements.

Our reply: The figure caption will be corrected as suggested and a map with locations will be included in the supplementary material. Regarding the statistical analyses, we have now conducted a Repeated Measures Analysis of Variance followed by a Tukey's post hoc test to test if the mineralization curves in Figure 3 are statistically significantly different. Often, the curves were significantly different despite a low absolute difference. We will therefore change the phrasing in the text a few places to make this clear. A section will be added to the Methods, explaining the type of analysis and conditions used. The result of the test will be added to Figure 3 in the form of letters denoting whether or not differences between soil types were found statistically significant.

Please also note the supplement to this comment:

https://www.biogeosciences-discuss.net/bg-2018-32/bg-2018-32-AC2-supplement.pdf